# A SPIKING SEQUENTIAL MODEL: RECURRENT LEAKY INTEGRATE-AND-FIRE

## ABSTRACT

Stemming from neuroscience, Spiking neural networks (SNNs), a brain-inspired neural network that is a versatile solution to fault-tolerant and energy efficient information processing pertains to the "event-driven" characteristic as the analogy of the behavior of biological neurons. However, they are inferior to artificial neural networks (ANNs) in real complicated tasks and only had it been achieved good results in rather simple applications. When ANNs usually being questioned about it expensive processing costs and lack of essential biological plausibility, the temporal characteristic of RNN-based architecture makes it suitable to incorporate SNN inside as imitating the transition of membrane potential through time, and a brain-inspired Recurrent Leaky Integrate-and-Fire (RLIF) model has been put forward to overcome a series of challenges, such as discrete binary output and dynamical trait. The experiment results show that our recurrent architecture has an ultra anti-interference ability and strictly follows the guideline of SNN that spike output through it is discrete. Furthermore, this architecture achieves a good result on neuromorphic datasets and can be extended to tasks like text summarization and video understanding.

## 1 INTRODUCTION

The terms of deep learning and the corresponding artificial neural networks (ANNs) derivatives have been dominating in subject of computer science and keep the current state-of-the-art performance in a widespread of machine learning's application scenario such as computer vision (Simonyan & Zisserman, 2014), natural language processing (Collobert & Weston, 2008), speech/audio recognition (Hinton et al., 2012), video understanding (Ye et al., 2015) since the first arising of the AlexNet (Krizhevsky et al., 2012), even some of them has beat the humans' cognitive level in certain tasks. However, ANNs fail to uptake the advantages of the Neuronal Dynamics, which instantiates as high-power consumption, relatively low responses and etc.

Spiking Neuron Networks(SNNs) (Maass, 1997), with inspiration for the propagation of the cortex neurons (Perrett et al., 1982; Tuckwell, 1988), have been presented continuous attention as a new, power-efficient and hardware friendly technology. In contrast to the mere implementation of spatial information and complicated float point computation of ANNs, SNNs utilize spatial-temporal dynamics to mimic the bio-behavior of neurons, as well as its dyadic-valued computation whose feeding electrical sequential impulses (i.e., spikes), belong to the binary-like set of $\{0,1\}$. Benefit from the capabilities of processing binary-spiking signal and consequential effectiveness, there is an alternative for SNNs that has a feasibility of further development of machine learning and neuromorphic application, which has been long-term significantly deployed in many neuromorphic hardware including SpiNNaker (Furber et al., 2014), TrueNorth (Akopyan et al., 2015) and Loihi (Davies et al., 2018).

In contrast to the ANNs' well advanced, salient, proficient training methodology that indicate the conception of BackPropagation(BP) (LeCun et al., 1998) along with its derivatives that consequently give rise to the convergence of ANNs and diverse categories of frameworks(ie. TensorFlow, PyTorch, et al.) that make it succinct and available to train more deeper networks. However, for one thing, there are not so much theoretically supported or potent procedure for tackling the issue of training SNNs, which limits SNNs from going deeper, therefore SNNs hardly fulfill the ability in real-world complex missions, such as video-based recognition/detection, natural language pro-

cessing et al.. For another thing, there no exit practical auxiliary frameworks that are capable to promote the mature structure of SNNs, which leads to the consequence of few application and rare forward-step development of SNNs.

There are still various efforts to make progress in training, deepening the depth and applications of SNNs, whereas many obstacles block the development of SNNs at the same time. As for training, there are many circumvention ways to strengthen the accuracy of SNNs, except for neuromorphic methodology such as spike-timing-dependent plasticity (STDP) (Serrano-Gotarredona et al., 2013), winner-taken all (WTA) (Makhzani & Frey, 2015). In the first alternative scheme, an ANN is trained firstly, then it is transformed into the SNN version whose network structure is the same as the above-mentioned ANN, and neurons analog the behavior of ANN neurons (Diehl et al., 2015). The other is the direct supervised learning, also called Gradient descend, which is a superior, prevalent optimization method for this learning procedure. In order to solve the issue of the non-differential problems of spikes, (Lee et al., 2016) proposed an alternate that treats membrane potential as differential signals and directly uses BP algorithm to train deep SNNs. To act as more bio-behavior, (Ponulak & Kasiński, 2010) introduced the remote supervised STDP-like rule to be capable of the learning of sequential output spike. Besides, (Urbanczik & Senn, 2009) proposed a novel leaning rule whose information will be embedded into the spatio-temporal information during learning of the spike signals. Nevertheless, most of the learning methods presented above are merely engaged in a single aspect of either spatial or temporal information. The applications started to spring up due to the incoming of the event-based cameras composed of Dynamic Visual Sensors(DVS) (Shi et al., 2018). The mechanism of DVS can be outlined as a simulation of the visual path structures and functionalities of the biological visual systems whose neurons asynchronously communicate and encode the visual information from environment as spatiotemporally sparse light intensity change in the form of spikes. On the strength of the event-based cameras, diverse event-based datasets were acquired such as Poker-DVS, MNIST-DVS (Serrano-Gotarredona & Linares-Barranco, 2015) and CIFAR10-DVS (Wu et al., 2019).

Embracing the event-based cameras and their derived datasets, a variety of monographs demonstrate the different methodologies whose intentions are to make a plausibility of the application of accordingly components. (Peng et al., 2016) proposed an event-based classification based on static learning method, named Bag of Events (BOE in short). This method denotes the events of corresponding to the activated pixel of the DVS as joint probability distribution. Moreover, this method tests on multiple datasets such as NMNIST, MNIST-DVS, Poker-DVS, and it reveals that BOE can significantly achieve competitive results in real-time for feature extraction and implementation time as well as the classification accuracy. (Neil & Liu, 2016) proposed a deep CNN to pre-process spiking data from DVS, which is used in various deep network architecture and is also used to achieve an accuracy of 97.40% on N-MNIST datasets, in spite of its complicated pre-processing approach. In terms of SNNs, (Indiveri et al., 2015) proposed a SNN architecture, named Feedforward SNN, which is based on spike-based learning and temporary learning, and it achieves 87.41% accuracy on MNIST-DVS datasets. (Stromatias et al., 2015) proposed a composite system, including convolutional SNNs, non-spiking fully connected classifier, and spiking output layer with its performance of 97.95% of accuracy.

Together with improving the performance and enhancing the convergence rate of SNNs, the goal that whether a method that can absorb both advantages of ANNs and SNNs can be achieved. To this end, we propose RLIF with both low computational complexity and biological plausibility, to explore its usage in real-world tasks. In summary, the major contributions of this paper can be listed as follows:

- We propose RLIF, which absorbs the biological traits from SNNs, follows the unroll structure of RNNs, and enables a seamless way to insert into any sequential model in common deep learning frameworks.

- A mass throughput can be implemented through the transition of binary information between an interlayer of RLIF and other sequential layers, which meets the basic principle that the emission of neuron trains are binary values. Furthermore, RLIF can be easily extended into neuromorphic chips since its peculiarity of hardware-friendly.

- The experiments conducted in general DVS-based datasets (MNIST-DVS, CIFAR10-DVS) and Chinese text summarization (LCSTS-2.0) show that our RLIF is capable of capturing key information through time and has lower parameters compared to its counterparts.

## 2 PREMISE OF UNDERSTANDING RLIF

As mentioned before, the core idea in our architecture is about how to absorb the biological traits of SNN into RNN. To this end, learning algorithm in SNN will be introduced first and then we do a simple analysis on basic LIF neuron model, which aims to highlight the most relevant parts to our RLIF.

### 2.1 LEARNING ALGORITHM FOR SNN

To the best of our knowledge, the learning algorithm for SNN could be divided into two categories: i) unsupervised learning algorithms represented by spike timing dependent plasticity (STDP) and ii) direct supervised learning algorithms represented by gradient-based backpropagation. Classical STDP and its reward-modulated variants (Legenstein et al., 2008; Frémaux & Gerstner, 2016), the typical SNN learning method which only use the local information to update the weights of model, surrender to difficulties in the convergence of models with many layers on complex datasets (Masquelier & Thorpe, 2007; Diehl & Cook, 2015; Tavanaei & Maida, 2016).

Illuminated by observing the huge success of backpropagation in ANN, researchers start to explore a new way about how can backpropagation be used in training SNN under the end-to-end paradigm. (Lee et al., 2016; Jin et al., 2018) have introduce spatial backpropagation method into training SNN which mainly based on conventional backpropagation. As to imitate the temporal characteristics of SNN, (Wu et al., 2018) pioneered the use of backpropagation in both spatial and temporal domains to train SNN directly, through which it achieved the state-of-the-art accuracy on MNIST and N-MNIST datasets. (Huh & Sejnowski, 2018) introduce a differentiable formulation of spiking dynamics and derive the exact gradient calculation to achieve this and (Neftci et al., 2019) use surrogate gradient methods to conquer the difficulties associated with the discontinuous nonlinearity. As a step further to increase the speed of training, (Wu et al., 2019) convert the leaky integrate-and-fire (LIF) model into an explicitly iterative version so as to train deep SNN with tens of times speedup under backpropagation through time (BPTT).

### 2.2 LIF NEURON MODEL

Leaky Integrateand-Fire (LIF) is the most common and simple model which can modeling neuron operations and some basic dynamic traits effectively with low computational costs. In general, we describe LIF neuron (layer $l$ and index $i$) in differential form as

$$\tau_{mem} \frac{\mathrm{d}U_i^l}{\mathrm{d}t} = -(U_i^l - U_{rest}) + RI_i^l \tag{1}$$

where $U_i$ refers to the membrane potential, $U_{rest}$ is the resting potential, $\tau_{mem}$ is the membrane time constant, $R$ is the input resistance, and $I_i$ is the input current (Gerstner et al., 2014). When the membrane voltage of neuron reaches it firing threshold $\vartheta$, spikes was released to communicate their output to other neurons. After each spike, $U_i$ is reset to the original resting potential $U_{rest}$.

since the input current is typically generated by synaptic currents triggered by the arrival of presynaptic spikes $S_j^l$, (Neftci et al., 2019) model the dynamics of operations during approximating the time course as an exponentially decaying current following each presynaptic spike by

$$\frac{\mathrm{d}I_i^l}{\mathrm{d}t} = \underbrace{-\frac{I_i^l}{\tau_{syn}}}_{\text{decay}} + \underbrace{\sum_j W_{ij}^l \cdot S_j^{l-1}}_{\text{feed\_forward}} + \underbrace{\sum_j V_{ij}^l \cdot S_j^l}_{\text{recurrent}} \tag{2}$$

Based on this, the simulation of single LIF neuron can be decomposed into solving two linear differential equations. As RNN, who accepts both the current input $x_t$ and the previously hidden state $h_{t-1}$ and updates the current state via non-linear activation function $\sigma(...)$, the basic form is

$$y_t = \sigma(W_x \cdot x_t + W_h \cdot h_{t-1} + b) \tag{3}$$

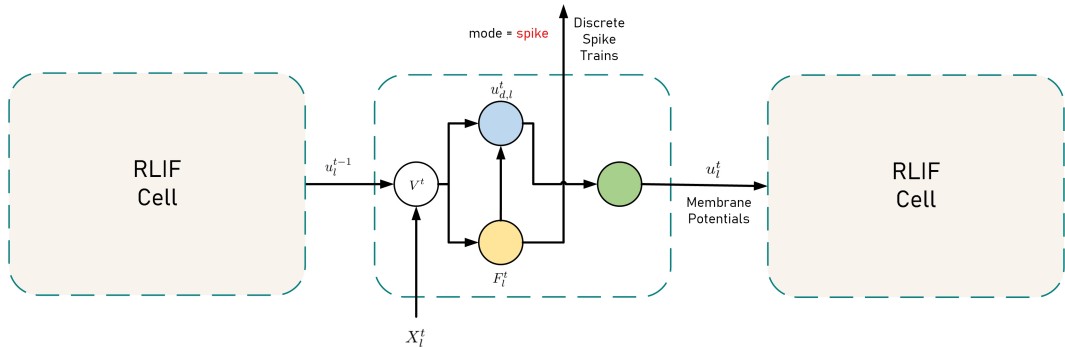

Figure 1: A diagram of RLIF cell.

Apparently, Equation 2 has the similar structure with basic RNN, which provides an insight about paraphrasing LIF into recurrent paradigm.

## 3 RLIF ARCHITECTURE

In this section, we will present the architecture of the Recurrent Leaky Integrate-and-Fire model (RLIF). The principle idea we hold is to enable RLIF with more biologically properties and achieve high computational efficiency meanwhile. As our architecture following by the paradigm of ANN, we treat the synaptic current of SNN as continuous probability distribution whereas keep the spike as discrete through a novel gradient broaden strategy, which allows the standard backpropagation through time in RLIF.

### 3.1 RLIF DEFINITION

Based on Equation 2 and 3 of LIF, we bring it into recurrent neural network's paradigm and the fusion form was described as follows:

$$V^t = U^t + u^{t-1} \tag{4}$$

$$F^t = V^t \geq V^t_{thres} \tag{5}$$

$$u^t_d = F^t \odot V^t_{reset} + !F^t \odot V^t \tag{6}$$

$$u^t = M^t + \boldsymbol{\beta} \tag{7}$$

where $V^t$ refers to the membrane potential with regard to the current voltage at timestep $t$ and recurrent membrane potential at timestep $t-1$, $F^t$ denotes whether the current voltage of neurons has reaches its own firing threshold $V^t_{reset}$, if it reaches its firing threshold then label this neuron with 1 otherwise 0. Next, we reset the firing neurons to its resting potential and let the membrane voltage of other neurons remain unchanged, as shown in Equation 6, the processed membrane potential $u^t_d$ are thus retrieved. Then we calculate $u^t$ with more biological plausibility as to mimic the random noise and accumulate with leakage.

$$Y^t_j = F^t \tag{8}$$

Here, the information firing between layers is $Y^t_j$ (binary output, as depicted in Figure 1, we denote this mode as `spike`). At current timestep $t$, where $X^t$ denotes the input, $U^t$ refers to the calculation of current voltage and $M^t$ represents the updating process of membrane potential. $V^t_{reset}$ is the reset voltage which produces the same effect (Lee et al., 2016) like $V_{reset}$ in Equation 6 as to simulate the inhibitory response of neurons and $V^t_{thres}$ is the firing threshold targeting at whether a neuron is fire or not. Besides, we propose two patterns in the calculation of $U^t$ and $M^t$: **FC** (short for Fully Connected) and **Conv** (short for Convolution):

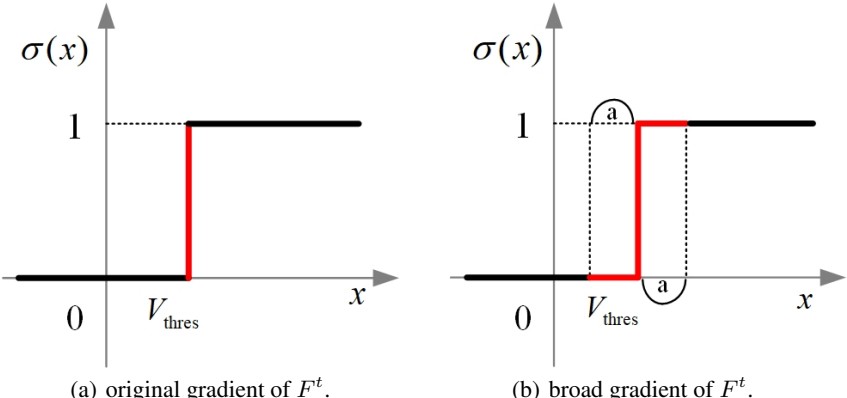

(a) original gradient of $F^t$.        (b) broad gradient of $F^t$.

Figure 2: The gradient of $F_t$ (red line refers to points that have gradient): (a) since $F_t$ is retrieved from a common step function, its gradient only exists in one point which leads to rather harsh conditions in updating learnable weights before it. (b) we use a novel gradient broading tactic to solve this problem with a hyperparameter $a$.

$$U^t = \begin{cases} W_{volt} \cdot X^t + b_{volt}, & \textbf{FC} \\ W_{volt} \otimes X^t + b_{volt}, & \textbf{Conv} \end{cases} \tag{9}$$

$$M^t = \begin{cases} \alpha \odot u_d^t + b_{mem}, & \textbf{FC} \\ \alpha \otimes u_d^t + b_{mem}, & \textbf{Conv} \end{cases} \tag{10}$$

where $\cdot$ represents matrix multiplication, $\odot$ refers to the hadamard product whereas $\otimes$ refers to the convolution product. $\alpha$ refers to the leakage to accumulate membrane potentials of each discrete timestep and $\beta$ is the mechanism of simulating random noise in mammal neurons.

As depicted in Figure 1, which is rather simple to further extend it into real-world complex tasks. Therefore, we set $V^t$ in Equation 4 as the hidden state $h^t$ of LSTM (Hochreiter & Schmidhuber, 1997) (as shown in Equation 11). Then followed by the same procedure as Equation 5 to Equation 8. The attention needs here is that we replace $h^t$ to membrane potential $u^t$ and keep the cell state $c^t$ unchanged. The usage of this RLIF variant (LIF-LSTM) will be introduced in the experiment of text summarization.

$$\begin{aligned} f^t &= \sigma(W_f \cdot X^t + U_f \cdot u^{t-1} + b_f) \\ i^t &= \sigma(W_i \cdot X^t + U_i \cdot u^{t-1} + b_i) \\ o^t &= \sigma(W_o \cdot X^t + U_o \cdot u^{t-1} + b_o) \\ c^t &= \tanh(W_c \cdot X^t + U_c \cdot u^{t-1} + b_c) \\ h^t &= o^t \odot \tanh(c^t) \end{aligned} \tag{11}$$

## 3.2 GRADIENT BROADING: broadgrad()

Since a critical problem has arisen due to the `spike` output mode: the nondifferentiable property of the binary spike trains output. Here, a rectangular function (Wu et al., 2018) $grad(...)$ was chosen to broaden the range of spike derivatives on the backward phase.

$$grad(F^t) = \begin{cases} 1, & [V_{thres}^t - a, V_{thres}^t + a] \\ 0, & other \end{cases} \tag{12}$$

As shown in Figure 2, hyperparameter $a$ is essential to determinate the range of $grad(F^t)$, which further exhibits considerable influence on the convergence of a network.

| Model | Method | MNIST-DVS | CIFAR10-DVS |
|---|---|---|---|
| (Zhao et al., 2014) | Composite system | 88.14% | - |
| (Stromatias et al., 2017) | Composite system | 97.95% | - |
| (Lagorce et al., 2016) | HOTS | - | 27.10% |
| (Shi et al., 2018) | Lightweight Statistical | 78.08% | 31.20% |
| (Paulun et al., 2018) | NeuCube | 92.03% | - |
| (Cannici et al., 2019) | Attention Mechanisms | - | 44.10% |
| (Sironi et al., 2018) | HATS | 98.40% | 52.40% |
| Ours | RLIF | **98.43**% | **56.93**% |

Table 1: The comparison of accuracy of RLIF and other methods on two neuromorphic datasets.

| Model | Total number of samples used | Acc |
|---|---|---|
| (Paulun et al., 2018) | 10.000 (scale-4) | 90.56% |
| (Paulun et al., 2018) | 10.000 (scale-8) | 92.03% |
| (Paulun et al., 2018) | 10.000 (scale-16) | 86.09% |
| Ours | 10.000 (scale-4) | 97.82% |
| Ours | 10.000 (scale-8) | **98.43**% |
| Ours | 10.000 (scale-16) | 92.46% |

Table 2: The comparison of the impact of different timesteps on MNIST-DVS.

## 4 EXPERIMENTS

We evaluate our proposed RLIF on image classification task to verify its effectiveness as compared with other brain-inspired methods. Moreover, we extend it into text summarization, a classical natural language processing task, the experiment shows that RLIF and its variant, with pluggability and flexibility inside, could be applied successfully into complex real-world tasks.

### 4.1 CLASSIFICATION ON NEUROMORPHIC DATASETS

Here, we used two neuromorphic datasets, MNIST-DVS and CIFAR10-DVS, with pixel resolution of 128 * 128 to verify the classification performance of RLIF. Both event-based datasets are taken from the original dataset by the DVS sensors, which in particular takes samples through moving along a fixed trajectory in front of the LCD monitor. MNIST-DVS dataset contains 30,000 event-stream records from handwritten digits 0 to 9, among which 80% are used for training and 20% for testing. The CIFAR10-DVS dataset contains 10 categories as well, totaling 10,000 event-stream records, each category containing 1,000 records.

DATA PREPARATION    The design of the pre-processing part is based on the statistical information of DVS data, which can effectively represent its temporal and spatial information. First of all, the pre-processing algorithm slides on the original event-streams sorted by timestamp according to a specific length event-window. The sliding step of the event-window is equal to its length. When the event-window slides, a new event-stream thus generated to represent the data with the same number of event-stream recordings as the event-window. Finally, each new event-stream is expanded into a three-dimensional data frame which we call event-frame. Therefore, an event-frame was retrieved by converting from a set of events, which represents the information of recording data at one timestep. After $\mathbf{T}$ times of processing, a record with timestep $\mathbf{T}$ can be obtained, which contains both spatial and temporal information of the original event-stream data, and its dimension is ($\mathbf{T}$, 128, 128, 2).

NETWORK STRUCTURE    As shown in Figure 3, the network receives the event-frame record of $\mathbf{T}$ timesteps followed by the pre-processing module and performs feature extraction. The addition of RIF in our network is the highlight, which serves as a key for high-efficient use of temporal information. The special layer (we denote it as `SumLayer`), which ultimately transformed the discrete binary event-stream into a continuous representation for overall prediction, through a way of integrating information of all timesteps.

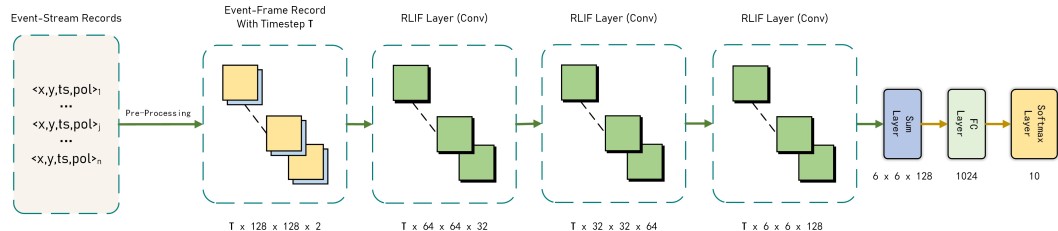

Figure 3: Our proposed network structure used in neuromophic dataset classification.

Most worthy of mention is, our model does not require complex pre-processing of the DVS raw event stream, and it can achieve better performance. Table 1 compares the performance of our model and the state-of-the-art methods in the MNIST-DVS and CIFAR10-DVS dataset, and our model achieves a relatively high accuracy in the test set. We obtained 98.43% accuracy on MNIST-DVS dataset, which is similar to the performance of ordinary convolutional network. Compared to the MNIST-DVS dataset, the CIFAR10-DVS dataset is more complex and contains much more information and noise than MNIST-DVS, but we ultimately achieved an accuracy of 56.93% that are better than all the SOTAs.

In order to verify the feasibility of the system, we compared the results of scale-4, scale-8, scale-16, in which the same preprocessing tactics were conducted on MNIST-DVS and trained with the same network model. The final test results are shown in Table 2.

## 4.2 TEXT SUMMARIZATION

Here, we proposed a sequence-to-sequence model (Seq2Seq) with LIF-LSTM (RLIF's variant) on LCSTS dataset.

DATASET    LCSTS is a large-scale Chinese short text summarization dataset, consisting of pairs of (short text, summary) collected by (Hu et al., 2015). The whole dataset, which consists of more than 2,400,000 pairs, was split into three parts under the same process as (Li et al., 2017; Ma et al., 2018) described. The noteworthy part is we only reserve pairs with scores no less than 3, thus we take PART I for training, filtered PART II for validation, and filtered PART III for testing. During our experiments, word segmentation was excluded whereas we only take Chinese character sequence as input.

EVALUATION METRIC    Here, we use the most common metric in evaluating the effect of text summarization: ROUGE score (Lin, 2004). The core idea of ROUGE is to compute the number of overlapping units between generated summaries and its reference summaries, including n-grams, word sequences, and word pairs. We use ROUGE-1 (unigram), ROUGE-2 (bi-gram) and ROUGE-L (LCS) as with previous exercises (Hu et al., 2015; Li et al., 2017; Ma et al., 2018) in the experimental results.

NETWORK STRUCTURE    As shown in Figure 4, our network is based on the sequence-to-sequence model where encoder is a stack of Layer Normalization (LN) and Bi-LSTM and decoder is similar to encoder of which Bi-LSTM is replaced by Uni-LIF-LSTM. In general, we use the final decoder layer and the final encoder layer output for obtaining the recurrent attention context through multi-head attention (Vaswani et al., 2017) and teacher-forcing strategy to supervise the learning of the representation of the source content with the corresponding summary. Under the assumption that word appears in the summary may existed in the text, a prior distribution is adopted here to make model prefer the word in text rather than others.

As the experiment result as depicted in Table 3, our LIF-LSTM appears to be a good substitute for LSTM but a step further of its biological plausibity, which demonstrates the feasibility of the usage of LIF-LSTM into real-world tasks.

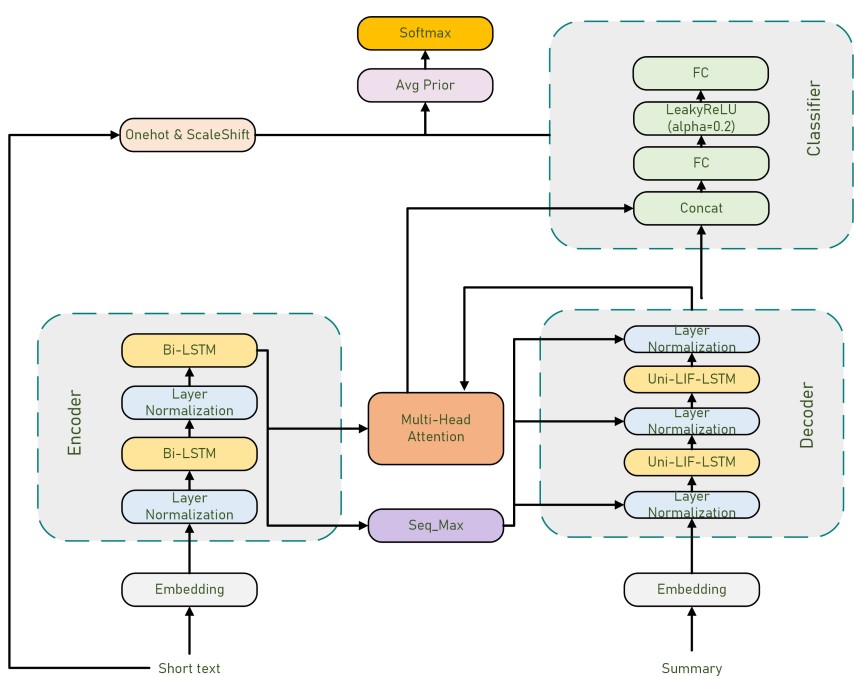

Figure 4: Model architecture for text summarization. The encoder on the left side has 2 blocks of LN and Bi-LSTM, the decoder on the bottom right, Uni-LIF-LSTM interspersed throughout LN. Common strategies like teacher-forcing, multi-head attention and the introduction of prior distribution have been taken to improve the performance as well as setting topK of beam search to 10.

| Model | R-1 | R-2 | R-L |
|---|---|---|---|
| RNN  (Hu et al., 2015) | 21.50 | 8.90 | 18.60 |
| RNN-context  (Hu et al., 2015) | 29.90 | 17.40 | 27.20 |
| SRB  (Ma et al., 2017) | 33.30 | 20.00 | 30.10 |
| CopyNet  (Gu et al., 2016) | 34.40 | 21.60 | 31.30 |
| DRGD  (Li et al., 2017) | 37.00 | 24.20 | 34.20 |
| Seq2Seq (superAE)  (Ma et al., 2018) | 39.20 | 26.00 | 36.20 |
| Ours | 37.22 | 24.64 | 34.45 |

Table 3: ROUGE-F1 on LCSTS test set (R-1, R-2, and R-L are short for ROUGE-1, ROUGE-2, and ROUGE-L, respectively).

## 5  CONCLUSION

In this paper, we propose the Recurrent Leaky Integrate-and-Fire (RLIF) model which has complementary advantages of ANNs and SNNs. RLIF is a more in-depth simulation on mammal neuron and can be easily plugged into the prevalent ANNs framework with the advantages of BPTT. The hybrid network of the combination of traditional ANNs module and RLIF converges easier than conventional SNNs method. The experiments show that our RLIF and its variant are of good application prospects due to their adaptability and stability, especially reflected in the text summary task. We believe that RLIF and its variant can be applied to many real-world challenging tasks such as neural machine translation, video understanding, which may lead to a shift in the public view about SNNs.

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

# A  SUMMARIZATION EXAMPLE OF OUR MODEL AND OTHER WORKS.

| |
|---|
| **Source**: |
| Last night, several people were caught to smoke on a flight of China United Airlines from Chendu to Beijing. Later the flight temporarily landed on Taiyuan Airport. Some passengers asked for a security check but were denied by the captain, which led to a collision between crew and passengers. |
| **Reference**: |
| Several people smoked on a flight which led to a collision between crew and passengers. |
| **Seq2Seq  (Ma et al., 2018)**: |
| China United Airlines exploded in the airport, leaving several people dead. |
| **Seq2Seq + superAE  (Ma et al., 2018)**: |
| Several people smoked on a flight from Chendu to Beijing, which led to a collision between crew and passengers. |
| **Ours**: |
| China United Airlines flight diverted to Chendu airport due to a smoking conflict. |

Table 4: The comparison of our LIF-LSTM model with  (Ma et al., 2018) on a text summarization example.

