# OpenReview forum: "A SPIKING SEQUENTIAL MODEL: RECURRENT LEAKY INTEGRATE-AND-FIRE"
_ICLR.cc/2020/Conference — Reject_

### Official Review · AnonReviewer2 · 2019-10-24
**Official Blind Review #2**

**Rating:** 3

**Review:**

This paper proposes a brain-inspired recurrent neural network architecture, named Recurrent Leaky Integrate-and-Fire (RLIF). Computationally, the model is designed to mimic how biological neurons behave, e.g. producing binary values. The hope is that this will allow such computational models to be easily implemented on neuromorphic chips and the solution will be more energy-efficient. On neuromorphic MNIST and CIFAR, the proposed model achieves higher classification accuracy than other listed methods. On ROGUE, a text summarization benchmark, the proposed model achieves competitive performance.

I am leaning towards rejecting this paper. The main advantage of the proposed computational model was not supported by evidence in the paper. The presented evidence only suggests that the computational model has the capacity for learning to solve real-world tasks to a degree that is on par with other existing computational models. But what supposedly distinguishes the proposed one from the rest, i.e. being more hardware-friendly and energy-efficient, was not demonstrated.

**Experience Assessment:**

I do not know much about this area.

**Review Assessment: Checking Correctness Of Derivations And Theory:**

I assessed the sensibility of the derivations and theory.

**Review Assessment: Checking Correctness Of Experiments:**

I assessed the sensibility of the experiments.

**Review Assessment: Thoroughness In Paper Reading:**

I made a quick assessment of this paper.

---

### Official Review · AnonReviewer1 · 2019-10-28
**Official Blind Review #1**

**Rating:** 1

**Review:**

####
A. Summarize what the paper claims to do/contribute. Be positive and generous.
####
The paper translates the Leaky Integrate and Fire model of neural computation via spike trains into a discrete-time RNN core similar to LSTM. The architecture would be readily amenable to the modern deep learning toolkit if not for the non-differentiability of the hard decision to spike or not. The hard decision is made by thresholding. The paper adopts a simple approximation of backpropagating a "gradient" of 1.0 through the operation if the threshold is within a neighbourhood [thresh - a, thresh + a], and otherwise 0.0, so the system can be trained by backpropagation.

The architecture is tested on a few "neuromorphic" video classification datasets including MNIST-DVS and CIFAR-DVS. Experiments are also run on a text summarization task.

####
B. Clearly state your decision (accept or reject) with one or two key reasons for this choice.
####

The reviewer thinks the paper should be rejected in its current state.

The proposed architecture is a straightforward change to a standard LSTM core. Thus it should be compared head-to-head to LSTM on standard datasets for these models (e.g. classic synthetic tasks, language modeling, speech recognition, machine translation, etc) with everything else held constant (hidden size, learning rate, sequence length, etc etc).

It also doesn't really carry over any of the benefits of Spiking Neural Nets even though it is inspired by Leaky Integrate and Fire because it operates in discrete time like a normal RNN, just with an extra binary output produced by spiking. It's unclear that a spiking inductive bias is actually useful, even though event-driven computation could in theory allow much less computation, the proposed method does not have that property.

So the paper doesn't really provide evidence to back up their claim that the proposed model combines the complimentary advantages of Deep Learning and Spiking Neural Nets.

####
C. Provide supporting arguments for the reasons for the decision.

While the proposed method is in-spirit inspired by the leaky integrate and fire model, it is operated/trained in discrete time which does not allow it to achieve the benefits of continuous time integrate-and-fire models which allow for less computation and time-discretization-invariance.

The conversion of the spiking model to the deep learning framework is rather crude, as the differentiable approximation to the non-differentiable threshold operation is biased and not well-motivated either empirically, intuitively, or theoretically (i.e. there are no comparisons to alternative choices).

There are new techniques for marrying continuous-time models and deep learning which seem more promising to investigate to this end (e.g. Neural ODE).

So in summary, the method doesn't have the computational benefits of a biologically plausible spiking algorithms and is not well-tested against competing deep learning methods, making it hard to verify the motivation of pushing toward a performant yet biologically plausible algorithm.
####

####
D. Provide additional feedback with the aim to improve the paper. Make it clear that these points are here to help, and not necessarily part of your decision assessment.
####
There are many grammatical and word-choice mistakes which make the paper hard to read.

Mainly, from a practical perspective, the paper would be much-improved by showing what benefit the spiking inductive bias confers over a standard LSTM on standard tasks in the deep learning community.

The method/landscape should be developed and studied in further detail until claims can be made about combining the strengths of spiking and deep-learning models.

**Experience Assessment:**

I have read many papers in this area.

**Review Assessment: Checking Correctness Of Derivations And Theory:**

I carefully checked the derivations and theory.

**Review Assessment: Checking Correctness Of Experiments:**

I assessed the sensibility of the experiments.

**Review Assessment: Thoroughness In Paper Reading:**

I made a quick assessment of this paper.

---

### Official Review · AnonReviewer4 · 2019-10-31
**Official Blind Review #4**

**Rating:** 1

**Review:**

Recently, it has been shown that spiking neural networks (SNN) can be trained efficiently, in a supervised manner, using backpropagation through time. Indeed, the most commonly used spiking neuron model, the leaky integrate-and-fire neuron (LIF), obeys a differential equation which can be approximated using discrete time steps, leading to a recurrent relation for the potential. The firing threshold causes a non-differentiability issue, but it can be overcome using a surrogate gradient. In practice, it means that SNNs can be trained on GPUs using standard deep learning frameworks such as PyTorch or TensorFlow.

Here the authors extend this approach by proposing two variations of the LIF model, called RLIF and LIF-LSTM. However, the presentation of these models is not clear at all.
For example:
* what is U^t in Equation 4?
* what is M^t in Equation 7?
* what is the difference (if any) between u^t and u_d^t?
Equation 8 is even more obscure. Why bothering defining a new variable Y if it is equal to F? What is index j, and why is it used only on the left hand side of the equation?

The description of the LIF-LSTM is even more obscure, nothing is defined.

Figure 2 has an error. On the left, with the heavyside activation function, the gradient is actually defined everywhere (with a value of 0) but on the red segment!!!

In addition, the experiments are not convincing. I am not an expert in NLP, so I will focus on the vision experiments.
Table 1 is incomplete. Wu et al 2019 (which they cite elsewhere!!!), reached 60.5% on DVS-CIFAR10, which is much better than this paper (56.93%)

For all these reasons, I recommend rejection.



**Experience Assessment:**

I have published in this field for several years.

**Review Assessment: Checking Correctness Of Derivations And Theory:**

I assessed the sensibility of the derivations and theory.

**Review Assessment: Checking Correctness Of Experiments:**

I assessed the sensibility of the experiments.

**Review Assessment: Thoroughness In Paper Reading:**

I read the paper at least twice and used my best judgement in assessing the paper.

---

### Decision · Program_Chairs · 2019-12-19

**Decision:**

Reject

**Comment:**

This work extends Leaky Integrate and Fire (LIF)  by proposing a recurrent version.
All reviewers agree that the work as submitted is way too preliminary. Prior art is missing many results, presentation is difficult to follow and incomplete and contains errors. Even if these concerns were addressed, the benefit of the proposed method is unclear. Authors have not responded.
We thus recommend rejection.